# Turning the tide against malaria in high-burden African countries: Trends, threats, and solutions

Loick Pradel Kojom Foko [1,2,3]*, Amit Sharma[3]

**1** Department of Animal Organisms, The University of Douala, Douala, Cameroon, **2** Center for Expertise and Research in Applied Biology (CEREBA), Douala, Cameroon, **3** Molecular Medicine/Structural Parasitology Group, International Centre for Genetic Engineering and Biotechnology (ICGEB), New Delhi, India

\* kojomloick@gmail.com

## Abstract

Despite decades of progress, malaria continues to impose an outsized health and economic burden in High Burden to High Impact (HBHI) countries. Using World Health Organization (WHO) data from 2015–2024, we analyzed epidemiological trajectories with the Mann-Kendall (MK) trend test, which handles heteroscedastic data, identified persistent bottlenecks, and assessed the effectiveness of current strategies. Actionable solutions have also been proposed for better control. Our analysis reveals sustained increases in malaria cases in five HBHI countries, namely Nigeria (MK statistics = 3.94, $p = 0.002$), the Democratic Republic of the Congo (MK statistics = 3.76, $p = 0.002$), Cameroon (MK statistics = 3.40, $p = 0.005$), Uganda (MK statistics = 2.86, $p = 0.02$), and Mali (MK statistics = 2.50, $p = 0.04$). Mortality trends remain heterogeneous and mostly non-significant across settings. These countries are facing the emerging of WHO biological threats to malaria control (antimalarial drug resistance, insecticide resistance, *pfhrp2* gene deletions, and *Anopheles stephensi* invasion), which intersect with silent challenges such as climate change, armed conflict, and population displacement, likely further complicating control efforts. Cabo Verde was recently certified malaria-free, underscoring that elimination remains attainable. A priority strategic recommendation emerging from this work is the strengthening of integrated, real-time malaria surveillance and response system, as well as the need to evaluate the impact of emerging hurdles (e.g., climate change). Achieving the WHO Global Technical Strategy target of ≥90% reduction in malaria burden by 2030 in HBHI countries is ambitious but feasible if locally driven, multisectoral solutions are implemented with urgency and sustained commitment.

## Introduction

Malaria has been documented in human history, with evidence of fatal infections in ancient Egypt and during World War I (1914–1918) [1]. The disease imposes

**Data availability statement:** All relevant data are within the paper and its Supporting Information files.

**Funding:** The author(s) received no specific funding for this work.

**Competing interests:** The authors have declared that no competing interests exist.

**Abbreviations:** 95%CI: Confidence interval at 95%, ACT: Artemisinin-based combination therapy, ART-R: Artemisinin partial resistance, AQ: Amodiaquine, AS: Artesunate, CHW: Community health worker, COVID-19: Coronavirus disease 2019, CQ: Chloroquine, DHA: Dihydroartemisinin, DRC: The Democratic Republic of the Congo, G6PD: Glucose-6-phosphate dehydrogenase, GMS: Greater Mekong subregion, GTS: Global Technical Strategy, HBHI: High Burden to High Impact, IM: Intramuscular, IPTp: Intermittent preventive treatment during pregnancy, IRS: Indoor residual spraying, ITNs: Insecticide-treated nets, LDMIs: Low Density Malaria Infections, LM: Light microscopy, n/a: Not applicable, NGS: Next-generation sequencing, NMCP: National Malaria Control Programme, Pf: Plasmodium falciparum, pfhrp2/3: Plasmodium falciparum histidine-rich proteins 2/3 genes, PoC: Plasmodium ovale curtisi, PoW: Plasmodium ovale wallikeri, Pk: Plasmodium knowlesi, Pv: Plasmodium vivax, PPQ: Piperaquine, PQ: Primaquine, RBM: Roll Back Malaria, RDT: Rapid diagnostic test, SMC: Seasonal Malaria Chemoprevention, SP: Sulfadoxine + Pyrimethamine, sSA: Sub-Sahara Africa, WHO: World Health Organization, WMR: World Malaria Report.

substantial health, economic, and social burdens in endemic areas [2]. It is caused by *Plasmodium* spp. parasites transmitted by infected female *Anopheles* mosquitoes, with *P. falciparum* (*Pf*), *P. vivax* (*Pv*), and *P. knowlesi* (*Pk*) driving most morbidity and mortality [2]; *P. malariae* (*Pm*), *P. ovale wallikeri* (*PoW*), and *P. ovale curtisi* (*PoC*) occur at lower incidence. Zoonotic species, such as *P. cynomolgi* and *P. inui*, have been reported in humans, particularly European travelers returning from Asia (e.g., Thailand or Cambodia) [3].

The World Health Organization (WHO) ranks malaria among the top global health priorities, alongside tuberculosis. In 2024, malaria caused an estimated 282 million cases and 610,000 deaths, with >90% in sub-Saharan Africa (sSA) [4]. Pregnant women, children under 5 years, and immunocompromised individuals are most at risk for severe outcomes and death [2]. Global control strategies include artemisinin-based combination therapies (ACTs), long-lasting insecticide-treated nets (LLINs), seasonal malaria chemoprevention (SMC), indoor residual spraying (IRS), and intermittent preventive treatment (IPT) in pregnancy and infancy [2], adapted to local epidemiology.

These interventions reduced global malaria incidence from 2000 to 2015, with a decline from 79.0 to 58.0 cases per 1,000 at-risk individuals, and mortality from 28.5 to 12.9 deaths per 100,000. Progress stalled after 2015, with global increases in cases [2]. WHO developed the Global Technical Strategy (GTS) for 2030 elimination, [2] targeting ≥40% reductions in morbidity and mortality by 2020, ≥ 75% by 2025, and ≥90% by 2030 versus 2015 [2]. The High Burden to High Impact (HBHI) initiative complements this by focusing on 11 high-burden countries [2].

Four HBHI countries—Nigeria (25.9% of cases, 30.9% of deaths), Democratic Republic of the Congo (DRC) (12.6%, 11.3%), and Uganda (3.0%, 5.9%)—contribute >40% of cases and ~50% of deaths [2]. Despite enhanced interventions (e.g., LLINs; SMC in Cameroon), burdens persist in HBHI settings (Fig 1) [2]. Only five African nations (Algeria, Cabo Verde, Egypt, Mauritius, Morocco) are WHO-certified malaria-free, highlighting the need to examine HBHI trends, control challenges, and solutions. This study analyzes 2015–2024 epidemiological data from HBHI countries, evaluates the limitations of existing strategies, identifies threats and gaps, and proposes targeted interventions.

## Materials and methods

### Study design

This study is a secondary analysis of routinely collected malaria surveillance data. The analyses were designed to describe temporal and spatial trends in malaria burden and related biological threats across High Burden to High Impact (HBHI) countries. The study was descriptive and exploratory in nature and was not intended to establish causal relationships between variables.

### Data sources

Country-level malaria data were obtained from publicly available World Health Organization (WHO) sources. Primary epidemiological data for the 11 HBHI countries

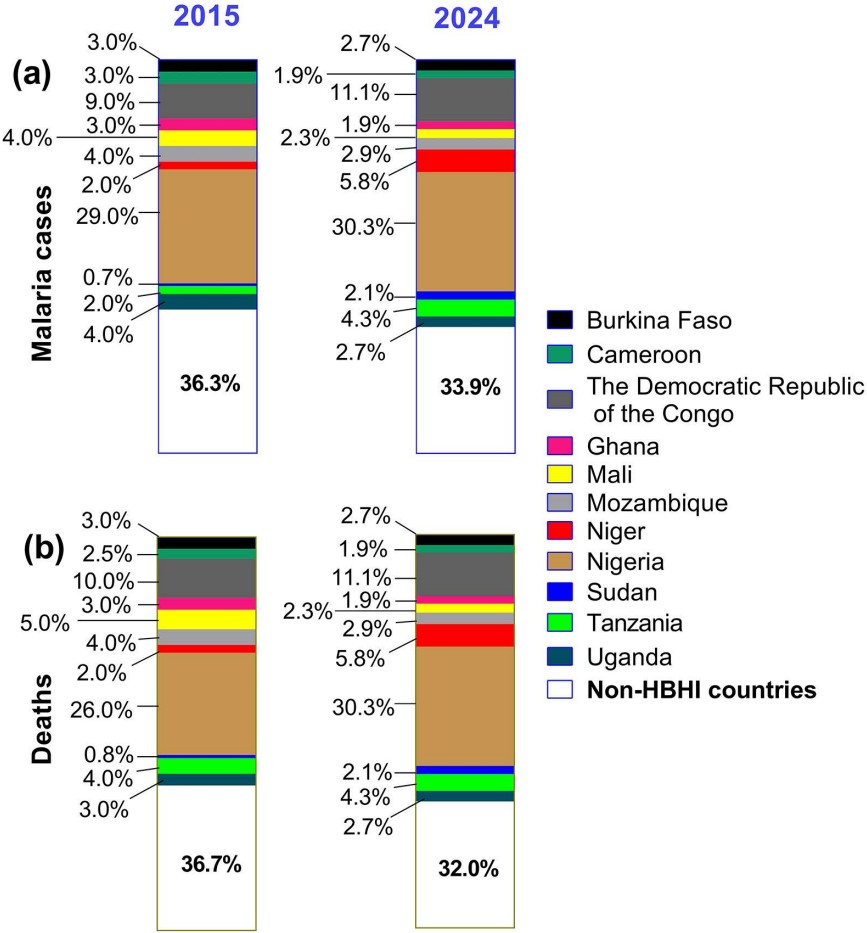

**Fig 1. Contribution of HBHI countries to global malaria cases (a) and deaths (b) in 2015 and 2024. Note.** HBHI: High burden to high impact country. The data were extracted from the WHO Malaria Reports 2016 and 2025. The graphs were generated using the GraphPad software v8.02 (GraphPad Software, CA, USA).

were extracted from successive editions of the World Malaria Report (2016–2025), which provide standardized annual estimates of malaria cases and deaths. WHO estimates incorporate surveillance, modelling, and adjustment methodologies. When inspecting malaria burden data from World Malaria Reports (WMR) 2016–2025, we have observed discrepancies in the same data from one report to another, which are due to WHO-made adjustments. Thus, for extracted data for the period 2015–2022, we have used the WMR 2023 data (S1 Table). For the data of years 2023 and 2024, the WMR 2024 and 2025 were used, respectively (S1 Table). Data were collated for the period 2015–2024 to enable consistent trend analysis across countries.

Information on national malaria control strategies implemented during the study period was abstracted from WHO reports and country profiles. Data on the current status of biological threats to malaria control and elimination—including antimalarial drug resistance, insecticide resistance, and diagnostic challenges—were retrieved from the WHO Malaria Threats Map (https://apps.who.int/malaria/maps/threats/). Where multiple updates were available, the most recent information within the study period was used, and data were aligned to the corresponding country-year as far as possible.

To contextualize WHO biological threats to malaria control in HBHI countries, i.e., antimalarial drug resistance and efficacy, *P. falciparum* histidine-rich protein 2 gene (*pfhrp2*) deletions, vector insecticide resistance, and *Anopheles stephensi* invasion,

and validate reported patterns, we conducted a structured literature review using PubMed. The search was restricted to English-language peer-reviewed articles and technical reports published between 2015 and 2024. Title and abstract screening were conducted to assess relevance to malaria control interventions and resistance patterns, followed by full-text review of eligible records. Searches were supplemented through expert consultation and backward reference mining of relevant publications. Only peer-reviewed articles and authoritative technical reports were included for critical appraisal and synthesis. The details of each search strategy, tailored to each WHO biological threat, are presented in S2 Table. For instance, the search strategy used for *pfhrp2* deletions is delineated as follows: ("Plasmodium falciparum"[MeSH Terms] OR "Plasmodium falciparum"[tiab]) AND ("histidine-rich protein 2"[tiab] OR "HRP2"[tiab] OR "pfhrp2 deletion"[tiab] OR "hrp2 gene deletion"[tiab]) AND ("gene deletion"[MeSH Terms] OR deletion*[tiab]) AND ("epidemiology"[Subheading] OR "prevalence"[tiab] OR "distribution"[tiab] OR "molecular surveillance"[tiab]) AND ("Africa South of the Sahara"[MeSH Terms] OR Africa[tiab] OR "Burkina Faso"[tiab] OR Cameroon[tiab] OR "Democratic Republic of the Congo"[tiab] OR DRC[tiab] OR Ghana[tiab] OR Mali[tiab] OR Mozambique[tiab] OR Niger[tiab] OR Nigeria[tiab] OR Sudan[tiab] OR Tanzania[tiab] OR Uganda[tiab]). No formal quality appraisal or meta-analysis was conducted, consistent with the narrative and contextual nature of the review.

## Data management and analysis

Extracted data were compiled into a harmonized database and cross-checked for internal consistency. Descriptive statistics were used to summarize annual malaria cases, deaths, and reported biological threats across countries and over time. Data visualization was conducted using GraphPad Prism v8.02 (GraphPad Software, CA, USA) to illustrate temporal trajectories and comparative patterns.

Temporal trends in annual malaria cases and deaths from 2015 to 2024 were assessed separately for each of the 11 HBHI countries using the Mann–Kendall (MK) trend test. The MK test is a non-parametric method appropriate for detecting monotonic trends in time-series data and is robust to handle non-normality and heteroscedasticity, which are common in surveillance datasets. Annual country-level counts of malaria cases and deaths were extracted and structured as univariate time series, with each observation corresponding to a single year. Analyses were conducted on available data without imputation; countries with incomplete time series were analyzed using all observed data points. The MK test assesses the presence of a monotonic trend by evaluating the relative ranking of observations over time and computing Kendall's tau ($\tau$), which quantifies both the direction and strength of the trend. Thus, a positive $\tau$ value indicates an increasing trend, whereas a negative $\tau$ value indicates a decreasing trend. Sen Slope corrections for autocorrelation were applied. The Theil–Sen slope method was used to estimate the slope (trend). This method is widely used to compute trends because it provides a robust and reliable estimate of slope that is not unduly influenced by outliers or non-normal data distributions [5]. Analyses were performed in R version 4.5 using the "trend" package. The mk.test() function was applied to each country-specific time series, with variance correction automatically implemented to account for tied values where present. To account for multiple comparisons in the Mann–Kendall test analyses, *p*-values were adjusted using the Benjamini–Hochberg procedure false discovery rate (FDR) correction method. Statistical significance was defined a priori as a two-sided $p\text{-value} < 0.05$ for all statistical analyses.

## Ethical considerations

This study utilized aggregated, publicly available secondary data with no individual-level identifiers. Therefore, ethical approval and informed consent were not required.

## Results

### A glimpse into demographic and eco-epidemiological features of HBHI countries

The HBHI countries show varied geographical locations, with nearly half of them located in the WHO West African region (Burkina Faso, Ghana, Mali, Niger, and Nigeria). In contrast, the rest of the countries are located in the Central African

region (Cameroon, DRC), East African region (Mozambique, Tanzania, Uganda), and Eastern Mediterranean region (Sudan). The countries are also distinct in terms of land size (from 0.238 million Km$^2$ for Ghana to 2.345 million Km$^2$ for Nigeria) and population size (from ~24 million inhabitants in Burkina Faso and Mali to ~224 million inhabitants in Nigeria) (Table 1). In general, HBHI countries are characterised by tropical and/or hot climates. *P. falciparum* (~100%) is the major

**Table 1. Geographical, demographic, and epidemiological facts about HBHI countries.**

| Characteristics | Land size (million km²) | Popula-tion size (million) | Climate | Malaria species | Mosquito species |
|---|---|---|---|---|---|
| **West Africa** | | | | | |
| **Burkina Faso** | 0.274 | 23.3 | Hot (Desert and semi-arid) and Tropical (Savanna) | *Pf* (~100%) and other species (*Pm*, *PoC*, *PoW*, *Pv*) (<1%) | *An. arabiensis*, *An. coluzzii*, *An. funestus*, *An. gambiae*, *An. melas*, *An. nili*, *An. pharoensis* and *An. stephensi*** |
| **Ghana**** | 0.238 | 34.1 | Tropical (Savanna) | | |
| **Mali** | 1.240 | 23.3 | Hot (Desert and semi-arid climate) and Tropical (Savanna) | | |
| **Niger** | 1.267 | 27.2 | Hot (Desert and semi-arid) | | |
| **Nigeria**** | 0.923 | 223.8 | Tropical (Monsoon and savanna climate) to Hot semi-arid | | |
| **Central Africa** | | | | | |
| **Cameroon** | 0.475 | 28.6 | Tropical (Rainforest, monsoon, savanna) to Hot semi-arid | *Pf* (~100%) and other species (*Pm*, *PoC*, *PoW*, *Pv*) (<1%) | *An. arabiensis*, *An. coluzzii*, *An. coustani*, *An. funestus*, *An. gambiae*, *An. melas*, *An. moucheti*, *An. nili*, *An. paludis* and *An. rufipes* |
| **The Democratic Republic of the Congo** | 2.345 | 102.3 | Tropical (Rainforest, monsoon, savanna) | | |
| **Eastern Africa** | | | | | |
| **Mozambique** | 0.801 | 33.9 | Mainly Tropical (Savanna) and Hot semi-arid | | |
| **Tanzania** | 0.945 | 67.4 | Mainly Tropical (Savanna) and Hot semi-arid | *Pf* mono-infection and mixed infections with *Pv* (99%), *Pv* (1%) and other (<1%) | *An. arabiensis*, *An. funestus s.l.*, *An. gambiae*, *An. leesoni*, *An. mascarensis*, *An. merus*, *An. nili*, *An. parensis*, *An. pharoensis*, *An. rivulorum*, and *An. rufipes* |
| **Uganda** | 0.241 | 48.6 | Tropical (Rainforest, monsoon and savanna) | | |
| **Eastern Mediterranean** | | | | | |
| **Sudan**** | 1.886 | 48.1 | Hot (Desert and semi-arid climate) | *Pf* and *Pv* (mainly as co-infections) | *An. arabiensis*, *An. culicifacies*, *An. d'thali*, *An. fluviatilis*, *An. funestus*, *An. hyrcanus s.l.*, *An. maculipennis*, *An. pulcherrimus*, *An. rhodesiensis*, *An. sacharovi*, *An. sergentii s.l.*, *An. stephensi***, *An. subpictus s.l.* and *An. superpictus* |

Note. Pf: *Plasmodium falciparum*, PoC: *Plasmodium ovale curtisi*, PoW: *Plasmodium ovale wallikeri*, Pv: *Plasmodium vivax*, Pm: *Plasmodium malariae*.

& *Plasmodium knowlesi* has not yet been reported in humans.

***Anopheles stephensi* has been reported in the country.

Data were sourced from the WHO Malaria Report 2024 (https://www.who.int/teams/global-malaria-programme/reports/world-malaria-report-2024), World Bank 2023 (https://datatopics.worldbank.org/world-development-indicators/), and Climate Knowledge Portal of the World Bank (Köppen-Geiger climate classification system) (https://climateknowledgeportal.worldbank.org/).

species involved in malaria cases and deaths, while non-*Pf* species collectively occurred at rates < 1%. In countries such as Uganda or Tanzania, malaria species are also represented by *Pf* and *Pf* + *Pv* (~99%), followed by *Pv* mono-infections (~1%), and non-*Pf/Pv* species (< 1%). The frequency of *Pv* infections is much higher in Sudan than that seen in countries like Tanzania. Emerging species such as *P. cynomolgi,* or widely prevalent in parts of Asia, such as *Pk,* have not yet been reported in the HBHI countries [3]. The vector mosquito spectrum in HBHI countries is rich, with a multitude of species, including *An. arabiensis*, *An. gambiae s.s.*, or *An. coluzzii*, which are primarily responsible for transmitting malaria parasites to humans (Table 1).

## Epidemiological trends of malaria statistics in HBHI countries

The malaria case trends during 2015 – 2024 for each HBHI country reveal contrasting patterns (Fig 2). Overall, morbidity cases have been on a constant rise from 2015 to 2024 for several countries, including Cameroon, DRC, Nigeria, and Uganda. Nigeria reported ~54.1 million cases in 2015 and ~68.5 million in 2024. In Cameroon, morbidity cases jumped from ~.9 million in 2015 to ~7.6 million in 2024, whereas in the DRC the cases increased from ~23.5 million in 2015 to ~37.2 million in 2024. This increase in malaria cases was even observed before the COVID-19 pandemic in 2020 in countries such as Burkina Faso, Cameroon, Uganda, and Nigeria (Fig 2). The highest increments in morbidity were observed in Sudan in 2024 (1.71-fold increase), Ghana in 2023 (1.23-fold increase), Uganda in 2016 (1.15-fold increase) and 2020 (1.12-fold increase), Cameroon in 2023 (1.14-fold increase), DRC in 2024 (1.12-fold increase), and Mali in 2020 (1.11-fold increase) (Fig 3). These observations were confirmed by the MK analyses. Indeed, five countries witnessed statistically significant increases in malaria across the years, namely Nigeria (MK statistics = 3.94, *p* = 0.002), DRC (MK statistics = 3.76, *p* = 0.002), Cameroon (MK statistics = 3.40, *p* = 0.005), Uganda (MK statistics = 2.86, *p* = 0.02), and Mali (MK statistics = 2.50, *p* = 0.04). For Burkina Faso, there was an increase in malaria cases, but the statistical significance was borderline (MK statistics = 2.33, *p* = 0.055) (Fig 4 & S1 Fig).

Regarding death cases, the temporal patterns have varied between countries, with four patterns: i) gradual increase followed by gradual decrease (inverted U-like) (DRC), ii) relative stagnation followed by an increase (Uganda, Tanzania, Niger, Nigeria, and Sudan), iii) gradual decrease followed by gradual increase (U-like) (Ghana, Mozambique), and jagged-like (Burkina Faso, Cameroon, Mali) (Fig 2). For instance, in Ghana, the number of deaths dropped from 12,800 in 2015–10,800 in 2022, then increased to 11,590 in 2024. In Nigeria, the number of deaths increased from 149,000 in 2015–184,830 in 2024 (Fig 2). The highest increments in death cases were observed in Sudan in 2023 (1.79-fold increase) and 2024 (1.92-fold increase) compared to years 2022 and 2023, respectively (Fig 5). The MK analysis revealed a statistically significant increase in death cases in Tanzania (MK statistics = 2.71, *p* = 0.03) (Fig 4 & S2 Fig). The remaining countries have not experienced a statistically significant decrease or increase in malaria deaths (Fig 4 & S1 Fig).

## Key malaria control strategies adopted by HBHI countries

Key strategies are variably implemented in the HBHI countries, except for ITNs, which are distributed freely at the national level via multiple channels such as antenatal care visits and nationwide campaigns (Table 2). In three countries (i.e., Cameroon, DRC, and Niger), the National Malaria Control Programme (NMCPs) do not recommend IRS., while in countries such as Burkina Faso, IRS is a cornerstone control strategy [6]. SMC is used in several countries with seasonal transmission, except in the DRC and Tanzania (Table 2). While malaria diagnosis is chargeable in private health facilities in these countries, the public sector in countries like Ghana or Mozambique provides free diagnosis of malaria cases, regardless of age. The majority of countries use ACTs to treat *Pf* malaria cases. Regarding *Pv* treatment, data are only relevant for two countries, i.e., Tanzania and Sudan, where *Pv* cases are more prevalent than in other countries. In both countries, primaquine (PQ) is recommended for radical cure of *Pv* cases, but the 'directly observed PQ-based therapy' strategy is only implemented in Zanzibar (Tanzania) (Table 2).

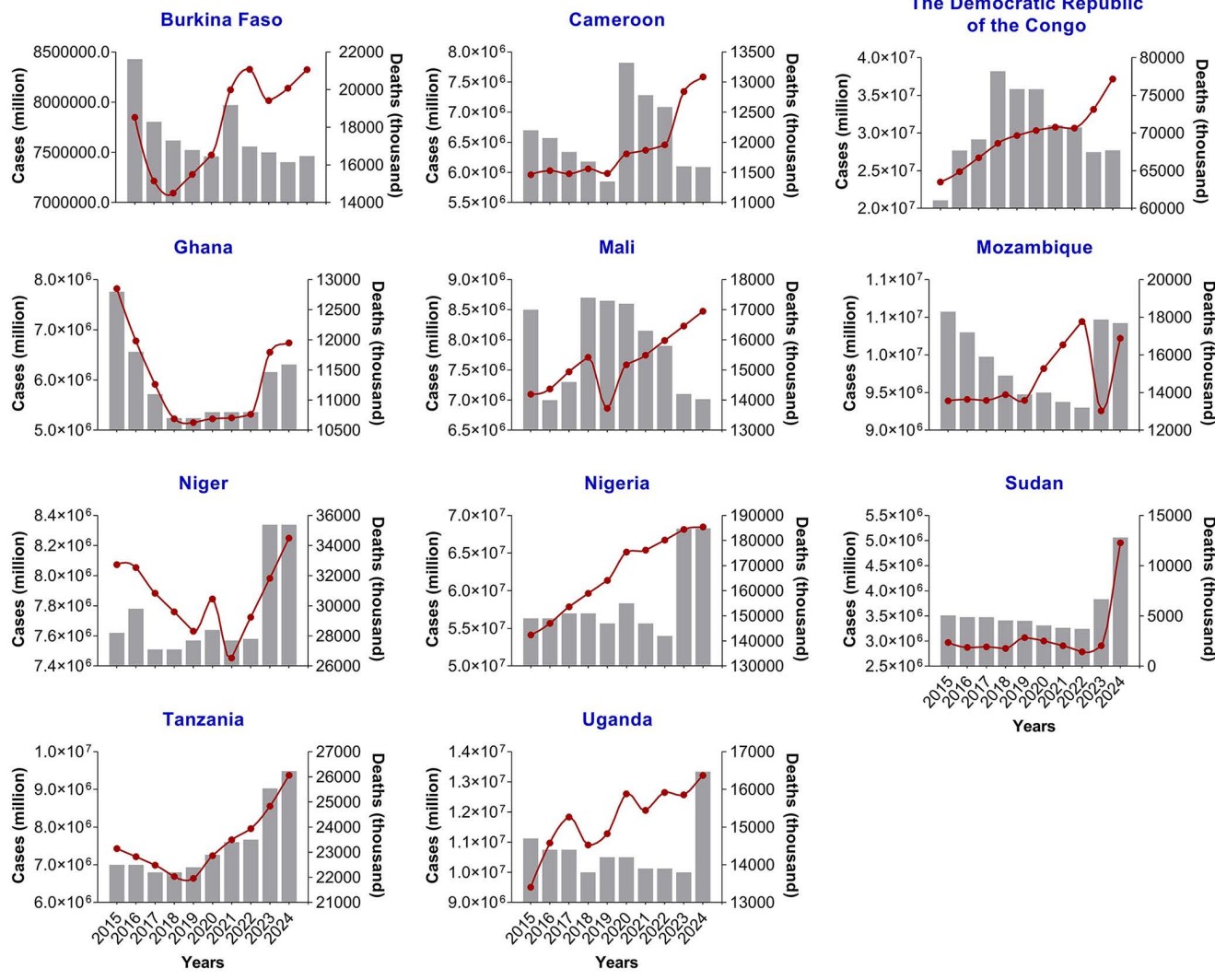

**Fig 2. Evolution of malaria cases and deaths in HBHI countries, 2015 – 2024. Note**. This figure illustrates the temporal evolution of malaria cases (red line) and deaths (grey bars), 2015 - 2024. The data were extracted from the WHO Malaria Reports 2023, 2024, and 2025. The graphs were generated using the GraphPad software v8.02.

## Current status of WHO biological threats to malaria control in HBHI countries

**Antimalarial drug resistance and efficacy.** The latest reports of antimalarial drug efficacy outline that commonly used ACTs, artemether + lumefantrine (AL), dihydroartemisinin + piperaquine (DHA + PPQ), or artesunate + amodiaquine (AS + AQ) are still highly efficacious for treating uncomplicated *Pf* malaria in HBHI countries. However, in the DRC, Moriarty *et al*. reported reduced corrected-PCR cure rates below the WHO threshold of 90%, for AS + AQ (73 – 100%), AL (86 – 98%), and DHA + PPQ (84–100%) [7]. The most commonly tracked genetic marker of artemisinin partial resistance is the *P. falciparum kelch13* (*pfk13*) gene. To date, a handful of mutations in *pfk13* (i.e., F446**I**, N458**Y**, C469**Y**, M476**I**, Y493**H**, R539**T**, I543**T**, P553**L**, R561**H**, P574**L**, C580**Y**, R622**I**, and A675**V**) have been strongly associated with either artemisinin partial resistance or treatment failure [8]. The validated *pfk13* mutations were reported in several HBHI countries (Fig 6). The mutation C580**Y**, frequently reported in the Greater Mekong subregion, was reported in two Cameroonian isolates [9].

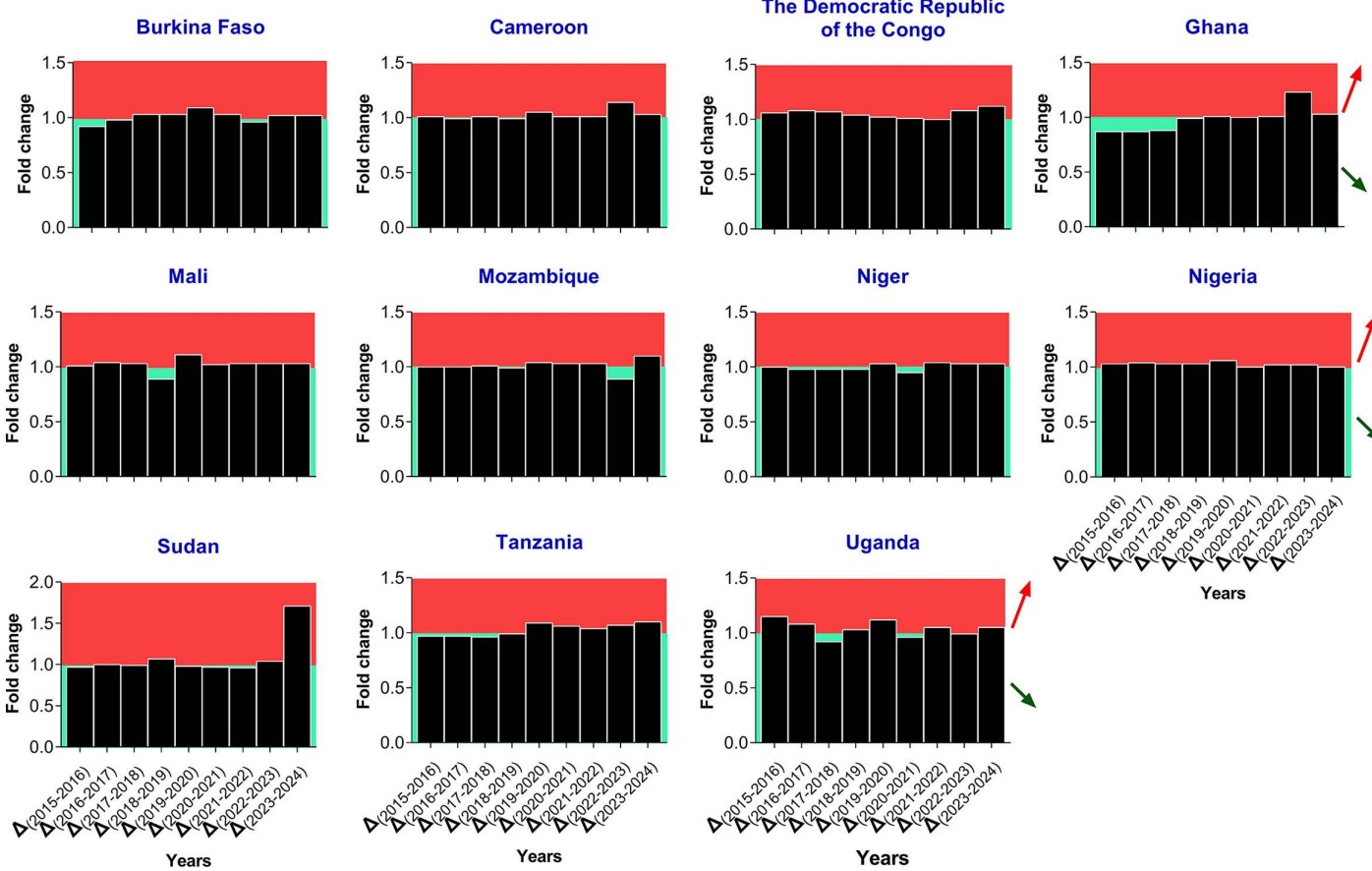

**Fig 3. Fold change of morbidity cases in HBHI countries, 2015 – 2024.** Note. The data were extracted from the WHO Malaria Reports 2023, 2024, and 2025. Fold change values between consecutive years were computed by dividing the number of cases of the year +1 by the number of cases of the year. Thus, fold change values above 1 outline an increase (red arrow, red plotting area), while values below the reference line or touching it mean a decrease (green arrow, green plotting area) or no change, respectively.

**Vector insecticide resistance.** The current situation of insecticide resistance in HBHI countries outlines high levels of insecticide resistance (0 – <90%) in all insecticide classes (https://apps.who.int/malaria/maps/threats/), with the exception of pyrroles that remain highly potent in experimental and real situations, even though some investigations reported a reduced susceptibility in *Anopheles gambiae* mosquitoes, collected in Ghana, DRC, and Cameroon, to the pyrrole insecticide chlorfenapyr [12]. In countries such as Burkina Faso, high resistance rates have been reported for five insecticide classes (i.e., carbamates, neonicotinoids, organochlorines, organophosphates, and pyrethroids) (Fig 6). In countries like Mali, Niger, DRC, Sudan, and Mozambique, high resistance rates have been reported against carbamates, organochlorines, and pyrethroids (Fig 6).

***Plasmodium falciparum hrp2* gene deletions.** Increasing reports of false negative rapid diagnostic tests (RDTs) results due to *hrp2* gene-deleted *Pf* parasites are publicly available [2]. Reports on the circulation of *pfhrp2* gene deletions have emerged from HBHI countries (https://apps.who.int/malaria/maps/threats/), except Niger, where, to the best of our knowledge, no studies have been conducted so far (Fig 6).

***Anopheles stephensi* invasion.** There are growing reports of the presence of *An. stephensi* in several regions in Africa (e.g., Nigeria, Sudan, Eritrea, Djibouti, Ethiopia, Somalia, Ghana, Kenya) and other South Asia countries (e.g., Sri

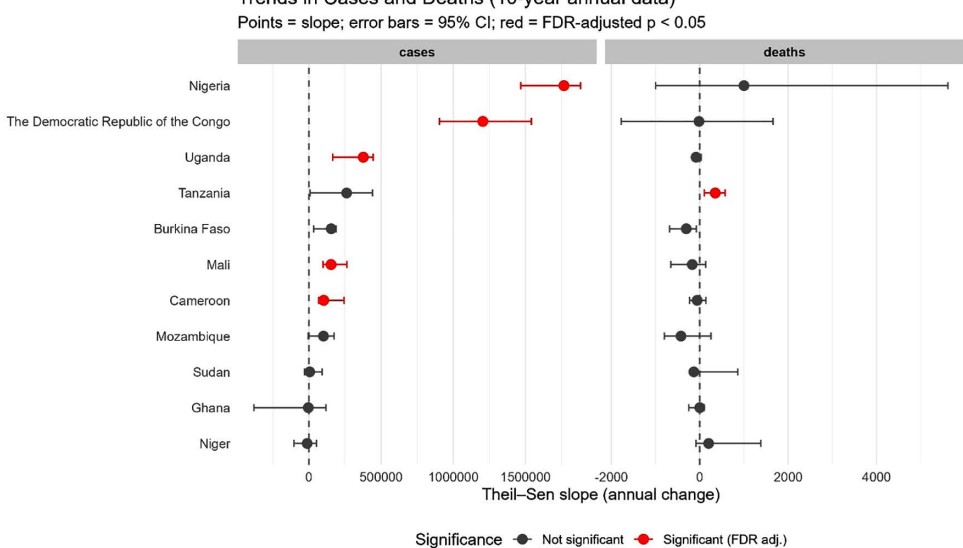

**Fig 4. Mann–Kendall trend analysis of cases and deaths of countries, 2015 – 2024.** Note. 95%CI: Confidence interval at 95%, FDR: False discovery rate. Mann-Kendall trend analysis was used to analyze the temporal evolution of malaria cases and deaths from 2015 to 2024. The Theil–Sen slope method was used to estimate the slope (trend). Statistical significance was set at $p$-value < 0.05.

Lanka) or Arabic countries (e.g., Yemen) [2]. Recent studies reported significant carriage rates of artemisinin-resistant and *pfhrp2*-deleted *Pf* parasites by *An. stephensi* populations from Dire Dawa, Ethiopia [13]. *Anopheles stephensi* invasion has been reported in several HBHI countries such as Nigeria (Awai, Kalkulum, and Kentengereng), Ghana (Tuba), and Sudan (e.g., Malliet, Sinkat, Sinnar City, Sola, or Tuli Island) (Fig 6) [2].

## The Cabo Verdean case: A recent success story of malaria elimination in sub-Saharan Africa

Cabo Verde is the first sSA country that managed to eliminate malaria from its territory. This archipelago (600,000 inhabitants, 4,033 km²) of 10 islands located on the West coast of Africa was officially declared malaria-free by the WHO on 12th January 2024, making this country the 44th country to benefit from this certification [2]. At this stage, preventing the re-establishment of malaria is the main challenge the country will face in the upcoming years. The successful elimination of malaria was not a smooth, steady process, as evidenced by the ups and downs of the 71 years of the fight [14]. The country has known two previous malaria elimination situations in 1967 and from 1983 to 1985, but the reintroduction of local cases and outbreaks wiped out these noticeable successes in the territory (Fig 7). Nevertheless, these two failures did not quench the Government's will to eliminate malaria. As of 2007, authorities had profoundly adjusted malaria-related health policies focusing on vector control, expanded diagnosis, and management of all malaria cases through early detection and rapid treatment. In addition, actions have been initiated to control the spread of malaria to neighbouring communities and to those at high risk of malaria resurgence or outbreak. Finally, control measures have been implemented to provide free diagnosis and treatment at airports and ports, aiming to prevent imported cases. Multisectoral collaborations, cross-border partnerships, population adherence to measures, and active surveillance have played a key role in these achievements. It seems that the development, implementation, and scale-up of these control strategies were the result of government-led efforts to invest more financial and human resources to achieve elimination. Cabo Verde's success highlights the importance of sustained political commitment, multisectoral collaboration, and active surveillance, which can serve as a model for HBHI countries facing similar challenges.

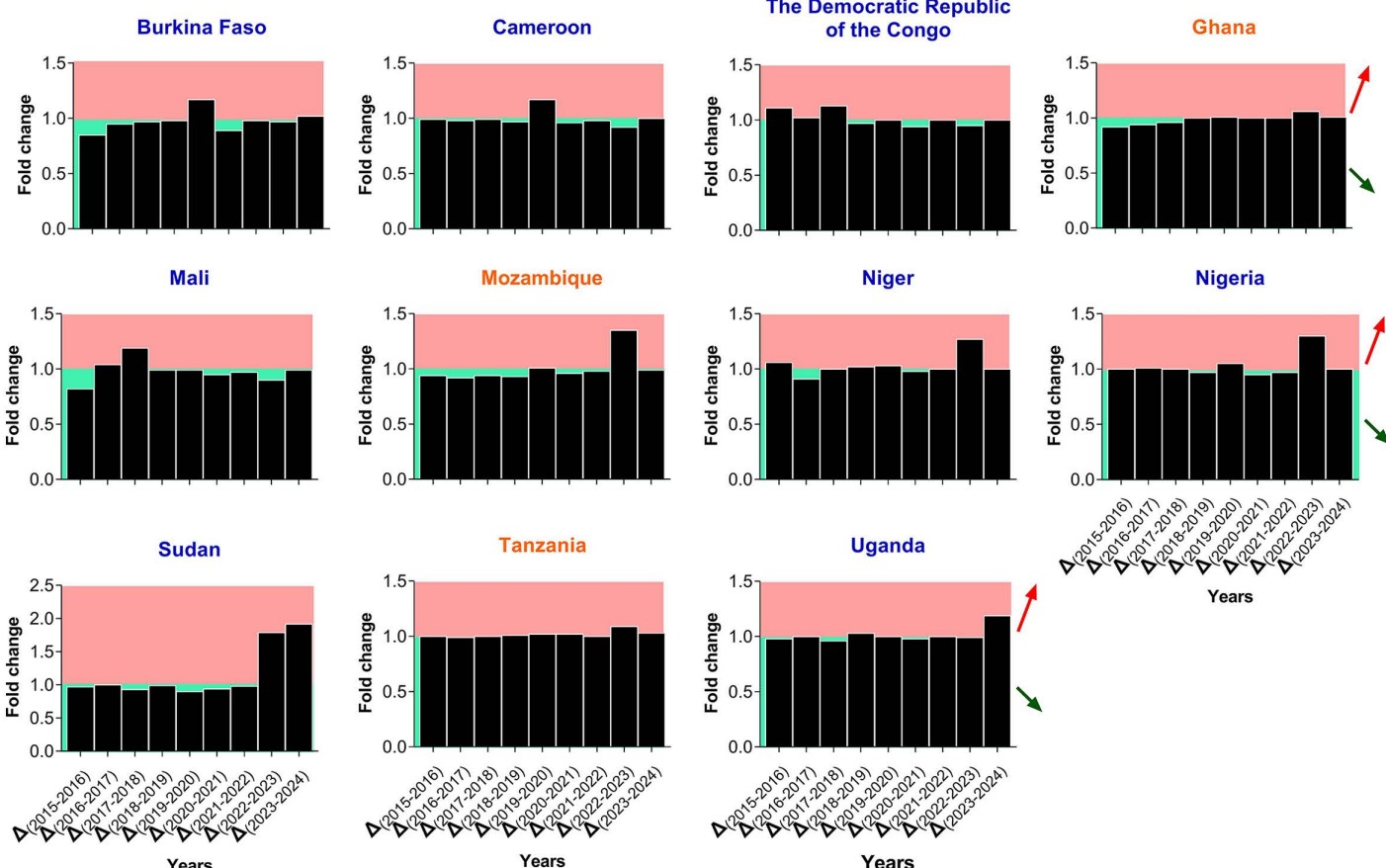

**Fig 5. Fold change of mortality cases in HBHI countries, 2015 – 2024.** Note. The data were extracted from the WHO Malaria Reports 2023, 2024, and 2025. Fold change values between consecutive years were computed by dividing the number of death cases of the year +1 by the number of death cases of the year. Thus, fold change values above 1 outline an increase (red arrow, red plotting area), while values below the reference line or touching it mean a decrease (green arrow, green plotting area) or no change, respectively.

## Discussion

Countries that achieved WHO malaria-free certification, e.g., China, have profoundly reinforced and expanded their malaria health systems. In HBHI countries, existing healthcare structures are generally resource-limited, even though advances in creating new healthcare facilities have been observed in some countries. Although new facilities exist, they are concentrated in urban areas, creating 'medical deserts' in rural/remote regions. The corollary is an unacceptable number of 'medical deserts' where neither health facilities nor competent healthcare givers are available or accessible [15]. Medical deserts are common in several HBHI countries and beyond. Populations in these areas are eager to indiscriminately resort to self-medication with drugs or herbal medicines [16], thereby exposing them to potential side effects and increasing the risk of severe complications and deaths. Also, these correctly diagnosed and treated 'forgotten' populations can constitute a significant reservoir for malaria parasite transmission. Finally, the strength of a healthcare service is reflected in its human resources, especially nurses and technicians. Thus, regular training and skill evaluation of human resources are fundamental for their continued accreditation and the optimal implementation of national and local malaria control strategies.

Countries such as Cabo Verde have expanded their national diagnosis and treatment networks to control malaria and prevent outbreaks efficiently [14]. However, when available at health facilities, diagnostic tools (i.e., RDTs and LM) are not

**Table 2. Key malaria control strategies in 'High Burden to High Impact' countries (World Malaria Report 2024).**

| Key control strategies | BF | CM | DRC | GH | ML | MZ | NE | NG[2] | UG | TZ[5] Mainland | Zanzibar | SU |
|---|---|---|---|---|---|---|---|---|---|---|---|---|
| **Insecticide-treated mosquito nets (ITNs)** | | | | | | | | | | | | |
| ITNs are distributed free of charge | yes[1] | yes[1] | yes[1] | yes[1] | yes[1] | yes[1] | yes[2] | yes[1] | yes[1] | yes[1] | yes[1] | yes[1] |
| ITNs are distributed through antenatal care visits | yes[2] | yes[1] | yes[1] | yes[1] | yes[1] | yes[1] | yes[2] | yes[1] | yes[1] | yes[1] | yes[1] | yes[2] |
| ITNs are distributed through the Immunization Programmes/baby clinics | yes[2] | yes[1] | yes[1] | yes[1] | yes[1] | no | yes[2] | yes[1] | yes[1] | yes[1] | yes[1] | yes[2] |
| ITNs are distributed via mass campaigns | yes[1] | yes[1] | yes[1] | yes[2] | yes[1] | yes[1] | yes[2] | yes[1] | yes[1] | yes[1] | yes[1] | yes[2] |
| **Indoor residual spraying (IRS)** | | | | | | | | | | | | |
| IRS is recommended by the National Malaria Control Programme | yes[2] | no | no | yes[1] | yes[2] | yes[1] | no | yes[2] | yes[1] | yes[2] | yes[1] | yes[2] |
| **Chemoprevention** | | | | | | | | | | | | |
| IPTp is used to prevent malaria in pregnancy | yes[1] | yes[1] | yes[1] | yes[1] | yes[1] | yes[1] | yes[1] | yes[1] | yes[1] | yes[1] | no | yes[2] |
| SMC is used | yes[1] | yes[1] | no | yes[1] | yes[1] | yes[1] | yes[1] | yes[1] | yes[1] | no | no | n/a |
| **Testing** | | | | | | | | | | | | |
| Malaria diagnosis is free of charge for all malaria cases (Private sector) | no | no | no | n/a | no | – | no | – | no | no | no | no |
| Malaria diagnosis is free of charge for all malaria cases (Public sector) | yes[3] | no | yes[3] | yes[4] | yes[4] | yes[4] | yes[3] | no | yes[4] | yes[3] | yes[4] | yes[3] |
| RDTs are used at the community level | yes[1] | yes[1] | yes[1] | yes[1] | yes[1] | yes[1] | yes[2] | yes[1] | yes[1] | yes[1] | yes[1] | yes[2] |
| G6PD test is recommended before treatment with PQ | n/a | n/a | n/a | n/a | n/a | n/a | n/a | n/a | n/a | n/a | no | no |
| **Treatment** | | | | | | | | | | | | |
| ACT for treatment of *Pf* malaria | yes[1] | yes[1] | yes[1] | yes[1] | yes[1] | yes[1] | yes[1] | yes[1] | yes[1] | yes[1] | yes[1] | yes[1] |
| Pre-referral treatment with quinine or artemether IM or artesunate suppositories | yes[2] | yes[1] | yes[2] | yes[2] | yes[1] | yes[2] | yes[1] | yes[2] | yes[2] | yes[2] | no | yes[2] |
| Single low dose of PQ with ACT to reduce the transmissibility of *Pf* | n/a | no | n/a | no | n/a | no | no | no | no | yes[2] | yes[1] | no |
| PQ is used for the radical treatment of *Pv* cases | n/a | n/a | n/a | n/a | n/a | n/a | n/a | n/a | n/a | n/a | yes[2] | yes[2] |
| Directly observed treatment with PQ is undertaken | n/a | n/a | n/a | n/a | n/a | n/a | n/a | n/a | n/a | n/a | yes[2] | no |

Note. Data were extracted from the WHO Malaria Report 2024 (https://www.who.int/teams/global-malaria-programme/reports/world-malaria-report-2024).

BF: Burkina Faso, CM: Cameroon, DRC: The Democratic Republic of the Congo, GH: Ghana, ML: Mali, MZ: Mozambique, NE: Niger, NG: Nigeria, UG: Uganda, TZ: Tanzania, SU: Sudan, ACT: Artemisinin-based combination therapy, G6PD: Glucose-6-Phosphate Dehydrogenase, IPTp: Intermittent preventive treatment during pregnancy, IRS: Indoor residual spraying, IM: Intramuscular, ITN: Insecticide-treated nets, n/a: Not applicable, *Pf*: *P. falciparum*, PQ: Primaquine, *Pv*: *P. vivax*, RDT: Rapid diagnostic test, SMC: Seasonal malaria chemoprevention.

1 The policy exists and has been implemented this year.

[2] The policy exists but has not been implemented this year or no data exists to support implementation.

[3] Only for RDT.

[4] Both for RDT and light microscopy.

[5] Where national data for the United Republic of Tanzania are unavailable, refer to Mainland and Zanzibar.

The sign '-'on the Testing section outlines that the question was not answered, and there is no information from previous years.

The mention 'n/a' (Not applicable) outlines that the strategy is not implemented in the country.

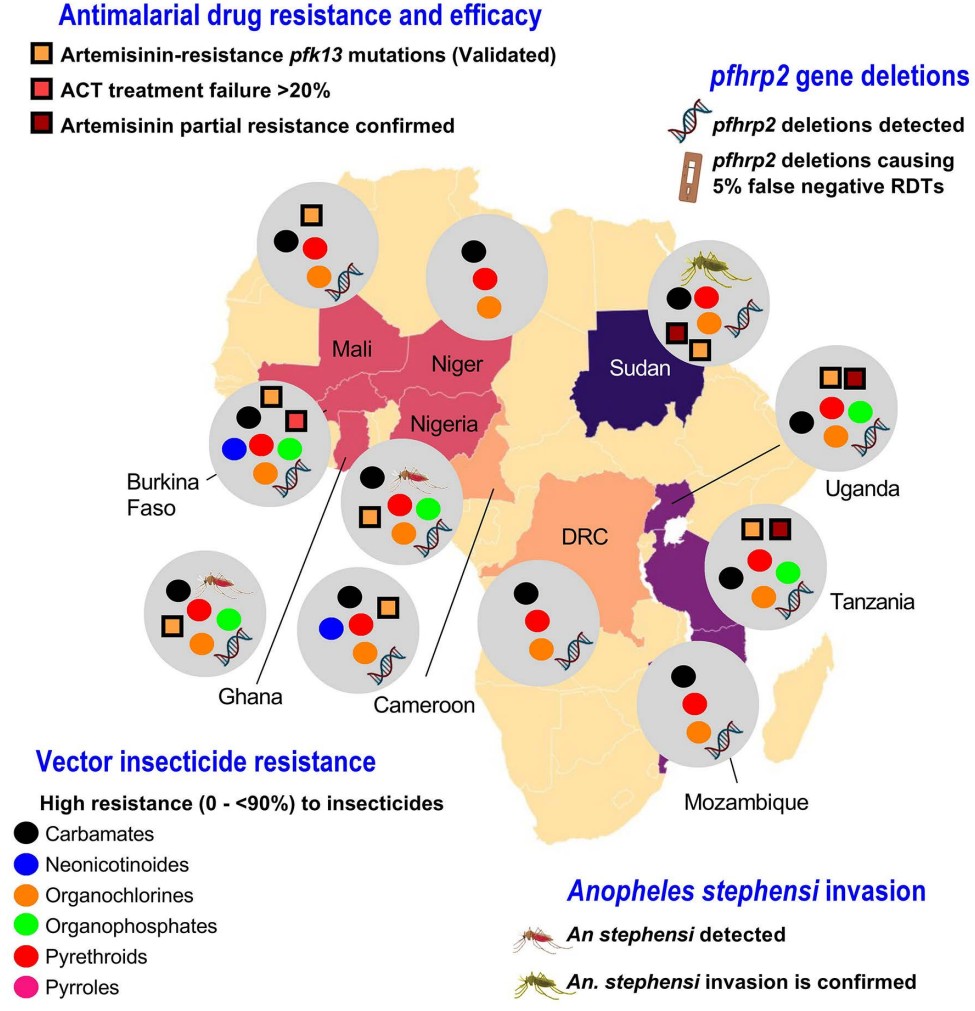

**Antimalarial drug resistance and efficacy**

- ☐ Artemisinin-resistance *pfk13* mutations (Validated)
- ☐ ACT treatment failure >20%
- ☐ Artemisinin partial resistance confirmed

***pfhrp2* gene deletions**

- *pfhrp2* deletions detected
- *pfhrp2* deletions causing 5% false negative RDTs

**Vector insecticide resistance**

**High resistance (0 - <90%) to insecticides**

- ● Carbamates
- ● Neonicotinoides
- ● Organochlorines
- ● Organophosphates
- ● Pyrethroids
- ● Pyrroles

***Anopheles stephensi* invasion**

- *An stephensi* detected
- *An. stephensi* invasion is confirmed

**Fig 6. Status of probable biological threats to malaria control as per WHO.** Note. ACT: Artemisinin-based combination therapy, pfhrp2: Plasmodium falciparum histidine-rich protein 2, *pfk13*: *Plasmodium falciparum* Kelch 13 gene, RDT: Rapid diagnostic test. The Africa continent basemap was downloaded from Natural Earth (Base layer of the map available from: http://www.naturalearthdata.com) and visualized using the QGIS software v3.36.1 (https://qgis.org/en/site/). The figure was generated using Microsoft PowerPoint and GraphPad software v8.02. The data were extracted from https://apps.who.int/malaria/maps/threats/ and [4,10,11].

consistently used. Innovative strategies involving automated RDTs could overcome this reluctance to routinely use the traditional diagnostic arsenal [17]. With the objective of drastically reducing malaria, it is crucial to streamline the diagnosis and treatment of malaria cases. Lastly, in several HBHI countries, preventive strategies (e.g., IPTp-SP) are often underused both at the regional and national levels. Regrettably, IPTp-SP coverage rates <50% are currently reported in HBHI countries such as Cameroon, Niger, and Mali [2].

All HBHI countries face the challenge of a shortage of sufficiently trained and competent workforce. Even though this shortage of workforce is also evident in near-to-elimination countries like India for malaria and other infectious diseases [18], the deficiency is likely higher in HBHI countries given their lower financial resources. Countries, e.g., China, Algeria, or Armenia, that managed to eliminate malaria from their territory have introduced a series of programmes to train and build the capacities of health workers [19]. Innovative approaches, such as blended learning, could be a new avenue for

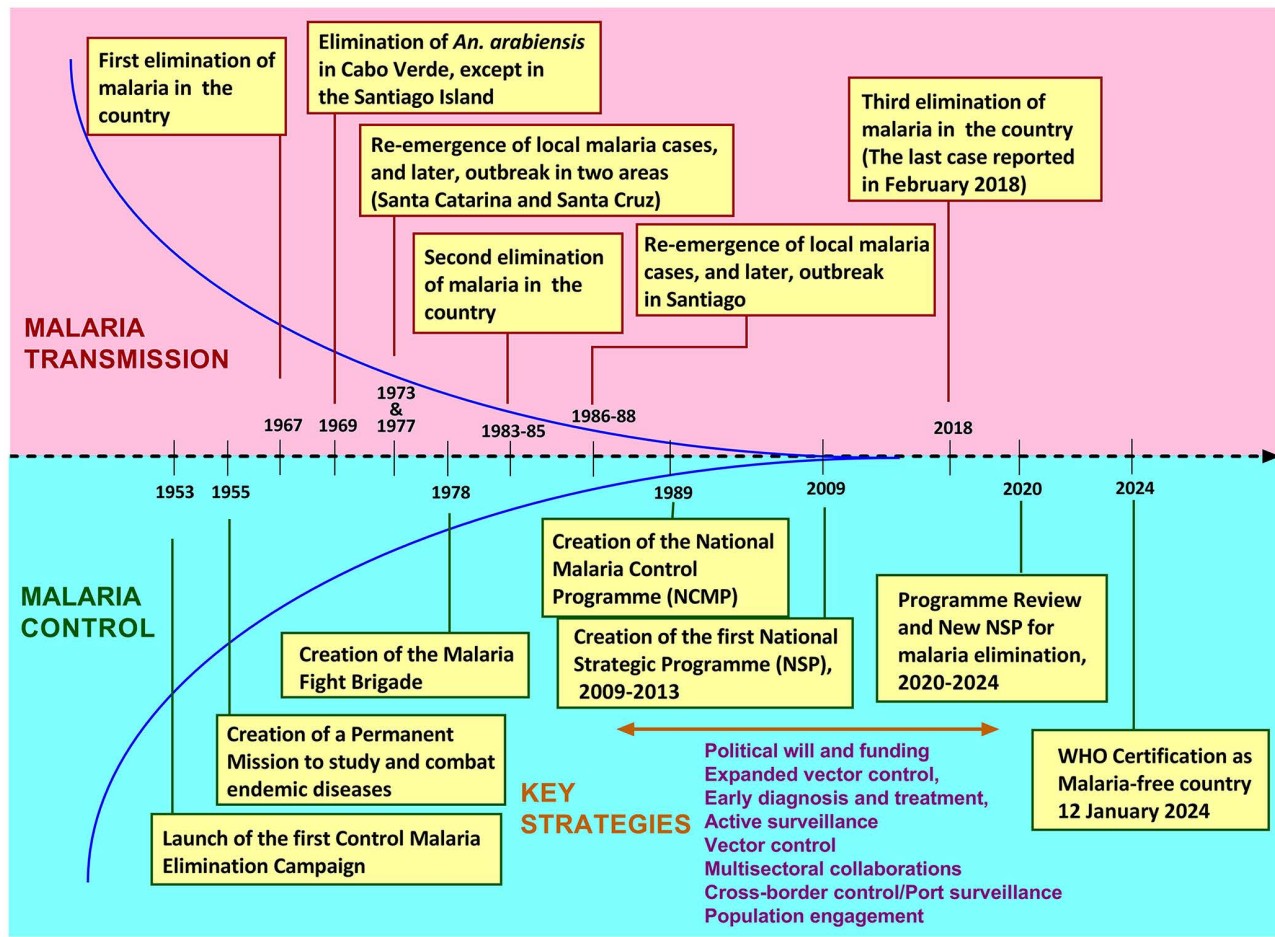

**Fig 7. Key moments in malaria elimination in Cabo Verde. Note.** The data were extracted from [2,14].

rapidly disposing developing a competent healthcare workforce as compared to the traditional lengthy face-to-face instruction [20]. Increasing and training the workforce is important, but training a knowledgeable workforce about their pivotal roles in population health is vital.

Community health workers (CHWs) are crucial to achieve effective and optimal delivery of healthcare services, particularly in hard-to-reach or remote populations. CHWs are more closely integrated into communities, as seen in several low- and middle-income countries, including HBHI countries (e.g., Mali). In practice, in countries such as Cameroon, malaria data are collected monthly to yearly basis and then collated for exploitation and analysis. This process is intrinsically lengthy, as it delays epidemiological analysis and implementation of tailored actions. In contrast, centralised near-real-time data-reporting digital systems, starting at the health facility level, could be a promising solution. NMCPs should have personnel disseminated in each health facility to collect and report malaria data in a timely manner once available. When a malaria case is confirmed, in-house NMCP officers can report malaria data on a digital platform (e.g., a malaria dashboard), allowing stakeholders (e.g., researchers, policymakers) to access updated data to evaluate the current epidemiological situation of malaria in a given region, thereby proactively implementing strategies.

Phenomenological analyses of control strategies are essential to implement them to their maximum potential in populations. New approaches, such as video-based health messages, evaluated among Ugandan pregnant women [21],

could be interesting adjunct approaches to accelerate malaria elimination efforts. The problem of community engagement is minimally double-faceted in HBHI countries. On one hand, studies reported low rates of awareness towards control measures (e.g., ITNs, IPTs) in countries such as Uganda, thereby limiting the utilization of these methods by populations [21]. On the other hand, reports of low acceptance or adherence, hesitancy, and reluctance to use prevention methods (e.g., ITNs, IPTs) often occur despite a high level of knowledge on the benefits of these methods. Some solutions, such as involving local and religious leaders, celebrities, or historical figures, could be fully leveraged to address these gaps in community engagement. This aspect is necessary to address future control strategies, such as the RTS, S and R21/ Matrix-M vaccines. Three HBHI countries (i.e., Ghana, Cameroon, and Burkina Faso) have now included the RTS, S vaccine, the leading malaria vaccine, in their national routine immunization policies [22]. Although comprehensive investigations into the extent and drivers of RTS,S vaccine hesitancy in countries like Cameroon have been conducted, systematic review-related findings outline a need to improve acceptance rates among Nigerian and Ghanaian mothers [23].

Creditable efforts have been made to provide updated data on the distribution and bionomics of malaria vectors in HBHI and non-HBHI countries via the continuous implementation of entomological studies [24]. It should be noted, however, that the national malaria vector maps, when available, give a rough idea of the distribution of mosquito malaria vectors at district levels at best. It is necessary to generate more granular data on the distribution of malaria vectors at the most precise geographical level (e.g., village) to determine whether tailored or standard vector control measures are necessary in areas within the same geographical limits.

It is time for HBHI countries to effectively implement and/or reinforce multisectoral collaborations to curtail malaria. Of course, the siloed approach is relatively faster due to minimal communication and reduced administrative bottlenecks; however, it is fragmentary and produces mitigated outcomes due to isolated efforts of different stakeholders. Also, this approach may often lead to irrational delivery of control measures. Interviewing several stakeholders in Ebonyi State, Nigeria, Omale *et al*. reported challenges such as the 'duplication in the provision of malaria commodities to health facilities by partners' [25]. Thus, it is indispensable for HBHI countries to leverage the evidence, benefits, and opportunities offered by intersectoral and multisectoral collaborations to achieve health equity for a malaria-free world.

Collaborative economic and political bodies, such as the Central African Economic and Monetary Community (CEMAC) in Central Africa and the Economic Community of West African States (ECOWAS) in West Africa, exist within the HBHI countries in the same geographical region. Regarding the fight against malaria, several regional institutions have been created to control and eliminate malaria. This achievement is more pronounced in the West African region, with a panoply of control entities, including Malaria Partners West Africa (MPWA), the Regional Centre for Vector Borne Diseases in West Africa (RCVBD), and the West Africa Integrated Vector Management Programme (WAIVM). In Central Africa, the scenario is more complex, with control actions mainly driven by external partners such as the WHO, the African Leaders Malaria Alliance (ALMA), and *Médecins Sans Frontières* (MSF). Africans are more inclined and legitimate to understand their realities and solve their problems. In this regard, it is crucial to leverage, implement, and/or reinforce regional collaborations through these entities to jointly coordinate transboundary control measures, effectively curbing malaria in the years to come.

Financial independence and national will are shared traits among countries that have eliminated malaria (e.g., Cabo Verde) or are near elimination (e.g., India). International funding boards typically have numerous concurrent engagements, and political, economic, and social factors can influence their funding preferences. Relying on its control actions over international funding can lead malaria-affected countries to accept disadvantageous compromises at the expense of their national interests. Several HBHI countries are still greatly dependent on the 'external help' to fight against malaria. In countries such as Nigeria, Mozambique, Sudan, and Tanzania, international funding accounted for >90% of the total budget allocated for tackling malaria [2]. Only DRC, Mali, and Cameroon strive to be less dependent on international funds, which account for ~15%, ~20%, and ~30% of the total budget allocated to malaria control, respectively [2]. The contribution of international help ranges between ~45 and 75% for the rest of countries. Financial independence could accelerate

the implementation of regular nationwide assessments of emerging and probable threats. This implies the need to fight against social defects such as corruption and fund mismanagement [26].

Alongside the above-mentioned requirements for efficient malaria control, HBHI countries should address the following understudied threats: armed conflicts, human movements, non-*Pf* species, low-density malaria infections (LDMIs), and climate change (Table 3).

A priority strategic recommendation emerging from this work is the strengthening of integrated, real-time malaria surveillance and response systems at the health-facility level. Given the demonstrated increase in malaria morbidity in several HBHI countries (e.g., Nigeria, DRC, Cameroon, Mali, and Uganda) and the heterogeneity of trends across settings, timely and granular data are essential to guide adaptive interventions. This could involve the deployment of digital reporting platforms (i.e., malaria dashboards) that enable near-real-time case notification, integration of routine diagnostic data, and systematic monitoring of WHO biological threats to malaria control. Such strategies should

**Table 3. Additional challenges to be addressed by HBHI countries for successful malaria control.**

| Challenges | Description |
|---|---|
| *Armed conflicts* | HBHI countries such as Cameroon, Nigeria, and Mali are facing armed conflict-driven political instabilities in some regions. Even though the impact of armed conflicts on malaria epidemiology is still elusive in these countries, consequences like serious hurdles for populations to access healthcare services in conflict-touched areas have been reported in Cameroon [27]. Also, armed conflicts force populations to flee towards relatively calmer regions, and this may negatively impact the capacity of health services in these regions to provide quality care to these internally displaced individuals [28]. |
| *Human movements* | The causes of human migration across internal and external borders are multiple, and not only the fact of armed conflicts. For religious or economic reasons, for instance, populations can move within and beyond national borders. Such movements can be the source for the emergence and/or spread of drug- or diagnostic-resistant parasites. A study outlined the risk of malaria transmission from walking pilgrims to surrounding populations living in Rajasthan, India [29]. |
| *Non-Pf species* | Non-*Pf* species, especially *Pv*, recent upsurges have been reported in endemic regions, such as the Eastern Mediterranean or South America [28]. There are rising reports on the occurrence of non-*Pf* species in HBHI countries, often at significant rates. In Cameroon, *Pv* has high rates among Duffy-negative individuals [30]. Similarly, the Tanzanian NMCP has reported significant *Pv* rates [2]. The control of non-*Pf* species, especially *Pv*, is trickier than that of *Pf* parasites, given the ability of *Pv* parasites to elicit dormant liver stages (hypnozoites) that cause relapses after the primary infection. Also, preventing *Pv* relapses requires PQ for killing hypnozoites, but this drug can provoke severe hemolysis in individuals with a deficiency in the glucose-6-phosphate dehydrogenase (G6PD). |
| *Low density malaria infections* | These infections are missed by usual diagnostic tools (RDT, LM) (submicroscopic/subpatent infections), and are generally seen in asymptomatic individuals. Low-density malaria infections (LDMIs) influence the epidemiology of malaria as individuals carrying such infections fuel parasite transmission [31]. In HBHI countries, LDMIs are highly prevalent. Nonetheless, their epidemiological and clinical impact has not been elucidated so far. Molecular techniques have played an instrumental role in the elimination of malaria in countries such as Sri Lanka and China [32]. Several authors argued the necessity to move towards molecular techniques (e.g., polymerase chain reaction) to address surveillance, diagnosis, and elimination of malaria, including LDMIs [33]. Some HBHI countries (e.g., Cameroon, Nigeria, Ghana, Tanzania) have comparable income levels to those of Sri Lanka (https://blogs.worldbank.org/en/opendata/world-bank-country-classifications-by-income-level-for-2024-2025). This fact underpins the possibility that these HBHI countries can afford and implement these techniques. |
| *Climate change* | Climate change is probably the most important global public health crisis, as it negatively impacts all living beings. Nearly 3.3 – 3.6 billion individuals, mainly living in Africa, the Americas, and Asia, are vulnerable to climate change [34]. Alarming signals of the negative impact of climate change on malaria epidemiology have been documented in Pakistan, where intense rainfalls and resulting flooding have exacerbated malaria cases in the Sindh and Balochistan regions [35]. Climate change correlates (e.g., rising temperatures, increased humidity, and rainfall) are associated with i) modification of the geographical distribution of malaria vectors that can move, for instance, towards historically malaria-free areas, and then provoke resurgences and outbreaks, and ii) reduced parasite incubation in mosquito vectors (e.g., *An. stephensi*) and then accelerate the production of ready-for-transmission parasites [36]. |

be coupled with decentralized decision-making capacity, allowing district-level health teams to rapidly adjust control strategies (e.g., targeted vector control). Although not directly assessed in this analysis, evidence from prior implementations suggests that strengthened surveillance systems can improve responsiveness and efficiency of malaria control programs, making them a critical component for accelerating progress toward control and elimination in HBHI settings.

## Limitations

A key limitation of this study is that this prospective analysis is based on absolute malaria case and death counts, which may be influenced by external factors not explicitly accounted for. Changes in population size and demographic structure across countries over the study period could affect the observed trends, as increases in cases may partly reflect population growth rather than true changes in transmission intensity. In addition, temporal improvements in surveillance systems, reporting completeness, and the adoption of more sensitive diagnostics may have contributed to apparent increases or fluctuations in reported cases and deaths. These factors should be considered when interpreting the observed temporal trends. Finally, fold changes should be interpreted cautiously because relative changes may appear large despite modest absolute differences.. Despite these considerations, the results still offer useful insights into multi-year temporal trends in malaria burden across diverse epidemiological settings.

## Concluding remarks

Here, we analyzed the epidemiological trends of the malaria burden in HBHI countries, identified knowledge and practice gaps, and proposed tailored solutions to control malaria in the upcoming years. The quantitative analysis demonstrates that malaria morbidity has increased significantly in five HBHI countries (Nigeria, DRC, Cameroon, Mali, and Uganda) between 2015 and 2024, while mortality trends remain heterogeneous and mostly non-significant across settings. Due to these observed rises in malaria morbidity in these HBHI countries and the varying trends across different settings, it is crucial to produce timely and detailed data for guiding adaptive interventions. Strengthened surveillance systems should be implemented to enhance responsiveness and efficiency in malaria control efforts.

## Supporting information

**S1 Table. World Malaria Report data of countries, 2015 – 2024.**
(DOCX)

**S2 Table. PubMed-based search strategies used to identify relevant papers on the status of WHO biological threats to malaria control in HBHI countries.**
(DOCX)

**S1 Fig. Mann–Kendall trend analysis of cases of countries, 2015 – 2024.**
(DOCX)

**S2 Fig. Mann–Kendall trend analysis of deaths of countries, 2015 – 2024.**
(DOCX)

## Acknowledgments

The authors acknowledge ICGEB (Trieste, Italy) for granting a prestigious career development Fellowship (F/CMR24–03) for senior researchers to the senior author Dr Loick P. Kojom Foko. We also thank ICGEB, New Delhi, India, for infrastructure support towards providing a working environment. We thank Mr. Aditya Chaki (Intern trainee, ICGEB, India) for the

help in downloading WHO data for generating Fig 2 and Table 2. Finally, the authors are grateful to Mr. Syed Shah Areeb Hussain (Project Research Associate III, ICGEB, India) for his inputs in statistical analyses.

## Author contributions

**Conceptualization:** Loick Pradel Kojom Foko, Amit Sharma.

**Data curation:** Loick Pradel Kojom Foko, Amit Sharma.

**Formal analysis:** Loick Pradel Kojom Foko.

**Investigation:** Loick Pradel Kojom Foko, Amit Sharma.

**Methodology:** Loick Pradel Kojom Foko, Amit Sharma.

**Project administration:** Loick Pradel Kojom Foko, Amit Sharma.

**Software:** Loick Pradel Kojom Foko.

**Supervision:** Amit Sharma.

**Validation:** Loick Pradel Kojom Foko, Amit Sharma.

**Visualization:** Loick Pradel Kojom Foko, Amit Sharma.

**Writing – original draft:** Loick Pradel Kojom Foko, Amit Sharma.

**Writing – review & editing:** Loick Pradel Kojom Foko, Amit Sharma.

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
