## [Decision Letter · Decision Letter 0]

13 Apr 2026

PGPH-D-26-00803

Turning the tide against malaria in high-burden African countries: Trends, threats, and solutions

Dear Dr. Kojom Foko,

Thank you for submitting your manuscript to PLOS Global Public Health. After careful consideration, we feel that it has merit but does not fully meet PLOS Global Public Health’s publication criteria as it currently stands. Therefore, we invite you to submit a revised version of the manuscript that addresses the points raised during the review process.

EDITOR: The manuscript has clear relevance but is presently limited by methodological weaknesses, insufficient analytical depth, and issues with data transparency. Major revisions are required to ensure the findings are robust, reproducible, and appropriately interpreted.

We look forward to receiving your revised manuscript.

Kind regards,

Kelechi Elizabeth Oladimeji

Guest Editor

Journal Requirements:

1. Please provide a complete Data Availability Statement in the submission form, ensuring you include all necessary access information or a reason for why you are unable to make your data freely accessible. If your research concerns only data provided within your submission, please write "All data are in the manuscript and/or supporting information files" as your Data Availability Statement.

For further assistance, you may go to: http://journals.plos.org/globalpublichealth/s/data-availability

2. Please provide separate main figure files in .tif or .eps format only and ensure that all files are under our size limit of 10MB.

3. Some material included in your submission may be copyrighted. According to PLOS’s copyright policy, authors who use figures or other material (e.g., graphics, clipart, maps) from another author or copyright holder must demonstrate or obtain permission to publish this material under the Creative Commons Attribution 4.0 International (CC BY 4.0) License used by PLOS journals. Please closely review the details of PLOS’s copyright requirements here: PLOS Licenses and Copyright. If you need to request permissions from a copyright holder, you may use PLOS's Copyright Content Permission form.

Potential Copyright Issues:

Figure 4: please (a) provide a direct link to the base layer of the map (i.e., the country or region border shape) and ensure this is also included in the figure legend; and (b) provide a link to the terms of use / license information for the base layer image or shapefile. We cannot publish proprietary or copyrighted maps (e.g. Google Maps, Mapquest) and the terms of use for your map base layer must be compatible with our CC-BY 4.0 license.

Additional Editor Comments (if provided):

Reviewers' comments:

Reviewer's Responses to Questions

**Comments to the Author**

1. Does this manuscript meet PLOS Global Public Health’s publication criteria? Is the manuscript technically sound, and do the data support the conclusions? The manuscript must describe methodologically and ethically rigorous research with conclusions that are appropriately drawn based on the data presented.

Reviewer #1: No

Reviewer #2: Partly

2. Has the statistical analysis been performed appropriately and rigorously?

Reviewer #1: No

Reviewer #2: I don't know

3. Have the authors made all data underlying the findings in their manuscript fully available (please refer to the Data Availability Statement at the start of the manuscript PDF file)?

Reviewer #1: No

Reviewer #2: Yes

4. Is the manuscript presented in an intelligible fashion and written in standard English?

Reviewer #1: No

Reviewer #2: Yes

5. Review Comments to the Author

Reviewer #1: This manuscript addresses an important and timely public health question and has potential value as a broad synthesis of malaria trends, threats, and strategic responses in high-burden African countries. The topic is clearly relevant to the readership of PLOS Global Public Health. However, in its current form, the manuscript requires major revision before it can be considered further. My main concerns relate to article framing, internal consistency of the results, adequacy of methodological detail, interpretation of the findings, data availability, and consistency between the text, figures, and tables.

Reviewer #2: The manuscript provides a timely and critical analysis of malaria epidemiology.

The article does not discuss confounders of absolute numbers like population growth or changes in notification and surveillance data and diagnostic tools. Neither does it recognize the small sample size as a limitation.

The authors should also provide the secondary analysis of the WHO data as a supplementary file.

6. PLOS authors have the option to publish the peer review history of their article (what does this mean?). If published, this will include your full peer review and any attached files.

**Do you want your identity to be public for this peer review?** For information about this choice, including consent withdrawal, please see our Privacy Policy.

Reviewer #1: **Yes:**Abubakari Abdul-wasid

Reviewer #2: No

Figure Resubmissions:

---

## [Editor Report · Decision Letter 1]

1 May 2026

Turning the tide against malaria in high-burden African countries: Trends, threats, and solutions

PGPH-D-26-00803R1

Dear Dr Kojom Foko,

We are pleased to inform you that your manuscript 'Turning the tide against malaria in high-burden African countries: Trends, threats, and solutions' has been provisionally accepted for publication in PLOS Global Public Health.

Best regards,

Kelechi Elizabeth Oladimeji

Guest Editor